# The Role of Beta-Endorphin in Food Deprivation-Mediated Increases in Food Intake and Binge-Eating

**DOI:** 10.3390/brainsci13020212

**Published:** 2023-01-27

**Authors:** Laica Tolentino, Asif Iqbal, Shafiqur Rahman, Kabirullah Lutfy

**Affiliations:** 1Graduate College of Biomedical Sciences, Western University of Health Sciences, Pomona, CA 91766, USA; 2Department of Pharmaceutical Sciences, College of Pharmacy, Western University of Health Sciences, Pomona, CA 91766, USA; 3Department of Pharmaceutical Sciences, University of Connecticut, 69 North Eagleville Road, Unit 3092, Storrs, CT 06269, USA; 4Department of Pharmaceutical Sciences, College of Pharmacy, South Dakota State University, Brookings, SD 57007, USA

**Keywords:** beta-endorphin, food deprivation, high-fat diet, binge-eating

## Abstract

Food deprivation and binge eating represent significant public health concerns. Previous studies have implicated that hypothalamic opioids are affected following food deprivation. However, the role of each opioid peptide is not fully understood. Therefore, we investigated the role of endogenous beta-endorphin in food deprivation-mediated increases in food intake and binge eating. Male mice lacking beta-endorphin and their respective controls were subjected to 24 h food deprivation and then were randomly assigned to receive a regular diet (RD) or a high-fat diet (HFD). After four to five weeks, animals were re-exposed to an HFD to assess if previous exposure to HFD would enhance binge-eating behavior. We report that food deprivation significantly increases food intake; however, beta-endorphin may not be involved in this process. In addition, our findings suggest that prior exposure to an HFD promotes binge-eating behavior in wildtype mice, and that these effects were modestly decreased in beta-endorphin knockout mice. Overall, our results support that beta-endorphin may play a modest role in mediating palatability-driven feeding, but not hunger-associated feeding. A better understanding of neural mechanisms involved in binge eating and deprivation-induced increases in food intake may inspire new prevention or treatment options to decrease the burden of eating disorders.

## 1. Introduction

Hyperphagia and food preference have been associated with food deprivation, as food is a natural reinforcer, and starvation or food deprivation can lead to an energy deficit and deprivation of rewarding stimuli [1,2]. Several studies have implicated the role of orexigenic neuropeptide Y (NPY) and agouti gene-related peptide (AgRP) in hyperphagia [3,4,5] and that the opioid system may be involved in this process [4,6,7]. For instance, opioid antagonists, such as naloxone and naltrexone, have been shown to suppress food intake in food-deprived animals, suggesting that the opioid system may be involved in fasting-induced food intake [8,9,10]. Furthermore, the endogenous opioid system has been implicated in mediating the rewarding aspects of feeding behavior [6,11]; however, the role of each opioid peptide in these behaviors is not fully understood.

Several studies have shown that food deprivation affects hypothalamic opioids, specifically the opioid peptide beta-endorphin. However, results reported in the literature have been variable depending on the duration of deprivation. While acute food deprivation (<48 h) increased beta-endorphin levels, chronic restriction decreased beta-endorphin levels. A 1983 study has initially shown that a 4-day food restriction increased beta-endorphin levels, while chronic food restriction for 16 days significantly decreased beta-endorphin levels in the arcuate nucleus [12]. On the other hand, Majeed and colleagues have shown increased hypothalamic beta-endorphin levels following 24 h food deprivation [13]. Consistent with these findings, a significant increase in the basal beta-endorphin release was observed from rat hypothalamic explants after depriving rats of food for 24 to 48 hours [14]. Furthermore, it has been noted that the mRNA level of beta-endorphin precursor, proopiomelanocortin (POMC), decreased following 24 h of food deprivation [15]. On the other hand, the POMC transcript is increased in mice following activity-based anorexia, a rodent model of anorexia nervosa [16]. Moreover, hypothalamic mu opioid receptor (MOP) gene expression significantly increased following food deprivation [4].

Based on the above findings, it is evident that endogenous beta-endorphin is affected following food deprivation or restriction; however, its role in feeding behavior under deprivation and binge eating is not fully understood. Our present study aimed to investigate further the role of endogenous beta-endorphin in food deprivation-mediated increases in food intake by utilizing a food deprivation paradigm. We examined changes in food intake after 24 h of food deprivation when the release of beta-endorphin is maximal [13]. To address this issue, we used genetically engineered mice unable to produce beta-endorphin [17], and their wildtype littermates. We also assessed whether this initial 24 h HFD exposure will lead to binge eating a month after the first exposure and whether beta-endorphin plays any functional role in this process. The selection of this time was based on our pilot studies showing that exposure to an HFD for 24 h leads to persistent binge eating (even when animals are tested a month later). We also selected this time to show that it is the neuronal plasticity causing binge eating, not the temporal vicinity of food exposure. We hypothesize that hyperphagia following 24 h of food deprivation will be reduced in mice lacking beta-endorphin compared to their wildtype littermates/controls. We also propose that this 24 h initial HFD exposure will induce persistent binge eating on an HFD even if animals are tested a month later, and beta-endorphin is involved in this enduring behavior.

## 2. Materials and Methods

### 2.1. Subject 

A total of 30 male beta-endorphin mice lacking the ability to produce beta-endorphin [17] and their wildtype littermates/age-matched controls (*n* = 6–9 mice per group), fully backcrossed for at least ten generations on a C57BL/6 mouse background bred in-house, were used for all experiments. All animals were acclimatized to the surroundings for one week with ad libitum access to a regular chow diet and water. Mice were individually housed in recyclable plastic cages with corncob bedding and maintained in a temperature-controlled room under a 12:12 h light-dark cycle (lights on 6 a.m./lights off 6 p.m.). All feeding experiments were conducted during the light cycle between 1 p.m. and 5 p.m. All experimental procedures involving animals were carried out with the approval of the Western University’s Institutional Animal Care and Use Committee (Pomona, CA, USA) and were acceptable according to the NIH guidelines for the care and use of animals in research. 

### 2.2. Diet 

The following diets were used for mouse experiments as indicated: regular chow diet (RD) (Envigo Teklad Global Diets 2018; 3.1 kcal/g with 18% fat, 24% kcal protein, 58% kcal carbohydrates), and high-fat diet (HFD) (Research D124292; 5.24 kcal/g with 60% fat, 20% kcal carbohydrates, and 20% kcal protein with blue dye). 

### 2.3. Food Deprivation Protocol

To determine how food deprivation affected food consumption between beta-endorphin knockout and their wildtype controls, mice were subjected to a 24 h deprivation. All measurements were conducted from 1 p.m. to 5 p.m. during the light cycle. Male beta-endorphin knockout mice and their wildtype littermates/age-matched controls were individually housed and habituated for one week with ad libitum access to RD and water. At the end of the acclimation period, baseline food intake was measured for four hours (0, 1, 2, and 4 h) and the following day (24 h later). Then, animals were randomly assigned to be deprived or not deprived for 24 h. For deprived animals, food was removed for 24 h with only access to water. Then, mice were assigned to be given RD or HFD for the next 24 hours, in which food intake was measured at 1, 2, 4, and 24 h. In addition, daily caloric intake was calculated using the following conversion: HFD = 5.24 kcal/g. At the end of day 3, all animals were restored with RD only. All animals were maintained and given ad libitum access to RD and water for 4–5 weeks before starting the binge protocol (Figure 1).

### 2.4. Binge-Eating Protocol 

Following four to five weeks of the last exposure to an HFD, animals were given access to HFD for one hour toward the end of the light cycle for one day. Food trays containing RD were removed 10 min before access to HFD. One-hour HFD intake was measured and recorded.

### 2.5. Statistical Analysis 

All data are represented as means ± SEM. Three-way ANOVAs with diet, time, and genotype as variables were used to analyze food and caloric consumption. Deprivation state, HFD exposure, and genotype were used for binge-eating results analysis. Significant interactions were followed up by Fisher’s Least Significant Difference (LSD) as the post hoc analysis. In addition, mixed effect analysis was used to analyze data with an unequal number of subjects. All statistical analyses were performed using GraphPad PRISMv.9.3.0 (GraphPad Prism, San Diego, California, USA). Statistical significance was set to *p* < 0.05.

## 3. Results

### 3.1. Food Intake Was Increased in Food-Deprived compared to Non-Deprived Mice Lacking Beta-Endorphin and Their Wildtype Controls 

Figure 2 illustrates food consumption (A) and caloric intake (B) at baseline (1, 2, 4, and 24 h) and after deprivation (49, 50, 52, and 72 h) in mice lacking beta-endorphin and their wildtype littermates. As shown in Figure 2A, food deprivation resulted in an increased food and caloric intake in both RD- and HFD-assigned groups (49–72 h) when compared to baseline measurements (1–24 h), as shown by a significant effect of time [F (7, 128) = 346.5; *p* < 0.0001; Figure 2A). All other main effects and interactions were not significant. The post hoc analysis indicated that food intake increased following deprivation in mice of both genotypes (compare the data for the 1 h vs. 49 h, 2 h vs. 50 h and 4 h vs. 52 h for each genotype), but there were no differences between beta-endorphin knockout mice and their wildtype controls.

Figure 2B depicts the caloric intake of deprived beta-endorphin mice compared to their wildtype littermates/controls. A three-way ANOVA revealed a significant interaction between time and diet [F (7, 128) = 37.68; *p* < 0.0001], and significant effects of time [F (7, 128) = 375.7; *p* < 0.0001] and diet [F (1, 128) = 105.7; *p* < 0.0001]. Post hoc analysis indicated that, under a deprivation state, mice fed an HFD consumed more calories than the RD-assigned group. Still, there were no differences between the mice of the two genotypes. 

We also evaluated the food (Figure 3A) and caloric (Figure 3B) intake of non-deprived mice. The HFD-assigned group consumed more calories when compared to the RD-assigned group at hours 49 (wildtype: *p* = 0.0425), 50 (wildtype: *p* = 0.0009; knockout: *p* = 0.0041), 52 (wildtype and knockout: *p* < 0.0001), and 72 (wildtype and knockout: *p* < 0.0001). A three-way ANOVA on food intake revealed a main effect of time [F (7, 80) = 312.9; *p* < 0.000], diet [F (1, 80) = 23.30; *p* < 0.0001], genotype [F (1, 80) = 6.64; *p* = 0.01], and significant interactions between time and food assignment [F (7, 80) = 5.216; *p* < 0.0001], and time and genotype [F (7, 80) = 2.39; *p* = 0.03]. Post hoc analysis indicated that beta-endorphin wildtype mice consumed more RD at 24 h (*p* < 0.0001) than the knockout mice assigned to the HFD group. However, this effect was absent between mice of the two genotypes assigned to the RD or HFD under the baseline condition (Figure 2, compare food intake for mice of the two genotypes at 1, 2, 4, and 24 h). In addition, food intake was significantly higher (*p* < 0.0001) in mice assigned to the HFD group than the RD group, regardless of genotype. Overall, these results suggest that animals consume more HFD than RD under both deprived and non-deprived states, but this response is not different between the mice of the two genotypes.

We also converted food consumption into calories, as shown in Figure 3B. A three-way ANOVA identified the significant effects of time [F (7, 80) = 323.7; *p* < 0.0001], diet [F (1, 80) = 129.0; *p* < 0.0001], and a time × diet interaction [F (7, 80) = 38.09; *p* < 0.0001], while all other interactions were not significant. Post hoc analysis revealed that the HFD-assigned group consumed more calories than the RD-assigned group (*p* < 0.0001; Figure 3B).

### 3.2. Comparison of Regular Diet vs. High-Fat Diet Intake in Deprived vs. Non-Deprived Mice Lacking Beta-Endorphin and Their Wildtype Controls

To assess the impact of deprivation on RD vs. HFD intake in mice lacking beta-endorphin and their wildtype littermates, we compared food intake in deprived (Figure 2) and non-deprived (Figure 3) groups at 49, 50, 52, and 72 h (Figure 4). A three-way repeated measure ANOVA of the data during the first hour (Figure 4A) revealed a significant effect of feeding state [F (1, 26) = 20.64; *p* = 0.001] and diet [F (1, 26) = 6.43; *p* = 0.02], but no significant effect of genotype (*p* > 0.05). There were no significant interactions between these factors. The post hoc test showed a significant increase in food intake in deprived compared to non-deprived wildtype mice, whether they were fed an RD (*p* = 0.001) or HFD (*p* = 0.004) during the first hour (49 h). Although the same increase in RD was observed in deprived than non-deprived mice lacking beta-endorphin (*p* = 0.003), the increase in HFD intake was insignificant (*p* = 0.20). 

Figure 4B shows food intake at the end of the second hour (50 h) in deprived and non-deprived wildtype and knockout mice. There were no genotype effects, but there was a significant effect of feeding state [F (1, 26) = 26.73; *p* < 0.0001] and diet [F (1, 26) = 23.35; *p* <0.0001] using three-way ANOVA. The post hoc analysis revealed increased HFD compared to RD intake in the non-deprived wildtype (*p* = 0.006) and knockout (*p* = 0.004) mice. In addition, an increase in RD intake was observed in the wildtype deprived group when compared to their non-deprived controls (*p* = 0.001). Likewise, HFD consumption was increased in deprived compared to non-deprived wildtype mice (*p* = 0.017). In contrast, intake of RD (*p* = 0.001) but not HFD (*p* > 0.05) was increased in deprived compared to non-deprived knockout mice. 

RD intake was higher in deprived than non-deprived mice of the two genotypes at 52h. Still, HFD consumption compared to RD intake was higher only in non-deprived mice of the two genotypes (Figure 4C). A three-way ANOVA revealed a significant effect of feeding state [F (1, 26) = 24.59; *p* < 0.0001] and diet [F (1, 26) = 28.79; *p* < 0.0001], but no significant interactions were found. The post hoc test identified a significant increase in HFD compared to RD intake in the non-deprived wildtype (*p* = 0.005) and knockout (*p* = 0.002) mice. In addition, the deprived group consumed significantly more RD when compared to their non-deprived controls in wildtype (*p* = 0.001) and knockout (*p* = 0.004) mice. However, these changes were not different between the mice of the two genotypes. Additionally, there was no significant increase in HFD intake in deprived compared to non-deprived mice of either genotype (*p* > 0.05).

Lastly, the food intake at the end of the 24 h testing period is shown in Figure 4D. A three-way ANOVA identified significant effects of feeding state (F (1, 26) = 14.63; *p* = 0.001) and diet (F (1, 26) = 16.29; *p* = 0.001), but no significant effects of genotype or interactions. The post hoc test revealed a significant increase in RD consumption in deprived compared to non-deprived wildtypes (*p* = 0.0158) and knockouts (*p* = 0.02). Furthermore, an increase in HFD compared to RD intake was seen in non-deprived wildtype (*p* = 0.03) and knockout (*p* = 0.002) mice, and this response was blunted in deprived mice of the two genotypes. Overall, these data suggest that deprivation increases the intake of both diets when food becomes available for a short term immediately after the 24 h deprivation period. Still, RD intake continued to increase for the entire 24 h test period in deprived compared to non-deprived mice. Yet, there is no difference in these responses between mice of the two genotypes. 

### 3.3. The Role of Beta-Endorphin in Binge Eating in Deprived and Non-Deprived Mice

To determine the role of beta-endorphin in binge eating, deprived and non-deprived mice were given access to an HFD for an hour, four to five weeks after the initial exposure to HFD. These are the mice we used for the deprivation and non-deprivation experiments described above. We used the first hour of exposure to HFD intake in the deprived (Figure 2, 49 h of RD-HFD) and non-deprived (Figure 3, 49 h of the RD-HFD) groups and compared them to the one we obtained 4–5 weeks later (Figure 5). A mixed-effect analysis indicated a significant effect of previous exposure to HFD [F (1, 32) = 12.13; *p* = 0.002] and a main interaction between exposure and deprivation state [F (1, 20) = 7.063; *p* = 0.02], while all other interactions and effects were not significant. Post hoc analysis showed that deprived wildtype mice consumed more HFD when compared to their non-deprived controls. However, this difference disappeared upon the subsequent exposure between mice of the two genotypes, showing that deprivation increased HFD intake, and thus binge-eating was not observed in deprived mice. Interestingly, non-deprived mice of either genotype increased their intake. Still, this effect was absent in the deprived group, suggesting that either they maximized their intake, or deprivation induced a reduction in binge eating.

## 4. Discussion

Food deprivation or restriction has been shown to enhance motivation for rewarding stimuli [1,18]. Several studies have demonstrated that mu opioid receptors (MOPs) are involved in hyperphagia and diet preferences that are associated with food deprivation. For example, significant reductions in deprivation-induced feeding were observed in mice following antisense probes directed against different exons of the MOP gene [7]. In addition, administration of MOP antagonist β-funaltrexamine (β-FNA) in deprived animals attenuated preference for HFD [4]. Since beta-endorphin is known as one of the endogenous ligands for MOP, we asked whether beta-endorphin regulates hunger-induced feeding following a 24 h food deprivation protocol. 

Our findings suggest that endogenous beta-endorphin is minimally involved in the deprivation-mediated increases in food intake in male mice regardless of diet type (RD or HFD) under the current experimental paradigm. While we demonstrated that food intake increases after food deprivation, we observed no significant differences in food and caloric intake in beta-endorphin knockout mice and their wildtype controls under a deprived condition. Our results are somewhat consistent with the study of Jewett and colleagues [19], in which they suggested a lack of opioid involvement in eating for hunger. Their study utilized a unique food deprivation paradigm in which they trained rats to discriminate between stimuli associated with 22h food deprivation (“hunger”) and 2h food deprivation (“satiation”), demonstrating that opioid antagonist naloxone could not reduce hunger-associated stimuli [19].

In the present study, we also investigated the impact of an initial 24 h HFD exposure in the presence and absence of deprivation on a subsequent (a month later) intake of HFD (for one hour). Based on our pilot studies, we proposed to assess whether this initial exposure to HFD would lead to binge eating. This response may be potentiated in animals experiencing food deprivation. Our results demonstrated that 24 h prior exposure to an HFD induced persistent binge eating in mice, as we observed that non-deprived mice exposed to an HFD displayed binge eating when tested for HFD intake 4–5 weeks after the first HFD exposure. To our knowledge, this is the first report to implicate that a single prior exposure to HFD promotes binge-eating behavior in non-deprived wildtype (*p* < 0.001) or knockout mice (*p* < 0.05) mice. The binge eating was modestly attenuated in beta-endorphin knockout mice compared to wildtype mice, as shown by a trend in the non-deprived mice. Nevertheless, our study may be underpowered; thus, more research is needed to delineate the role of beta-endorphin in this response in detail in the future.

Interestingly, as opposed to our hypothesis, we observed no further increases in HFD intake in deprived mice with prior HFD exposure. We believe that the lack of binge eating in this group is possibly due to the maximal amount of food they eat, as they already have a very high intake of HFD upon the first exposure after food deprivation. Although several studies have shown that MOPs regulate palatable food intake [6,20], the effects on food deprivation are still poorly characterized. Our finding demonstrated that previous exposure to an HFD enhanced binge-eating behavior, and that there was a modest decrease in food consumption in mice lacking the ability to express the peptide. Consistent with these findings, Hayward and colleagues had previously examined that mice lacking beta-endorphin or enkephalin peptides had decreased motivation to acquire food reinforcers during a non-deprived state, suggesting that the hedonic value of food predominates over the energy state [21]. On the other hand, the loss of beta-endorphin may lead to compensatory changes in enkephalin and dynorphin. In addition, the lack of endogenous peptides may lead to differential regulation of opioid receptors. Therefore, future studies involving mice lacking other endogenous opioid peptides (i.e., double knockout mice lacking beta-endorphin and enkephalin) would be of interest for further elucidation.

Overall, our current findings determine a modest role of endogenous beta-endorphin in binge eating behavior, as confirmed in the non-deprived mice. There was a trend toward reduction in binge eating in beta-endorphin knockout mice compared to their wildtype littermates, and we might have missed an opportunity to observe a difference between the mice of the two genotypes. This may be due to a smaller sample size used in the current study. It may have been possible to observe a difference between the mice of the two genotypes if we had increased the sample size. Yet, animal welfare issues need to be considered.

We did not see a significant difference in food consumption following deprivation between beta-endorphin deficient mice and their wildtype littermates/controls. It should be noted that several factors, including the length of deprivation and protocol design, may affect these results. As previously stated, beta-endorphin levels vary depending on the length of food deprivation or restriction. Several studies have shown an increased level of beta-endorphin following a deprivation [14,22], while others observed decreased levels of the opioid peptide [12,23]. Therefore, the lack of involvement in beta-endorphin in food intake observed in our findings may be due to the acute deprivation (24 h) protocol. Thus, a possibility exists that changes in food consumption may be observed in animals deprived after a longer period of deprivation. Consistent with this notion, a 48 h deprivation protocol has increased MOP expression, while changes were not seen at earlier time points [4]. Considering that we conducted our studies during the light phase of the animals, the results could have been different if these experiments had been conducted during the dark cycle since animals eat more during their dark cycle. However, given we did not observe much of an impact of beta-endorphin after deprivation, when animals eat more, we may not observe a difference. Nevertheless, further studies are needed to test whether any changes would be observed if these experiments were conducted during the animals’ dark cycle.

In addition, given that we used only male mice in the current study, it is possible to observe a sexual dimorphism in these responses. This notion is based on the observations that estrogen receptors (αERs) are expressed on POMC neurons, and that POMC mRNA fluctuates throughout the different phases of the estrus cycle. Ovariectomy has been shown to decrease POMC mRNA [24]. Furthermore, knocking out the αER on POMC neurons leads to increased food consumption and body weight in female but not male mice [24]. Notably, beta-endorphin levels are increased in female but not male mice following activity-based anorexia, a rodent model of anorexia nervosa [16]. Thus, it is possible to observe male/female differences, and possibly greater differences between wildtype and knockout mice. However, further studies are needed to test these possibilities.

## Figures and Tables

**Figure 1 brainsci-13-00212-f001:**
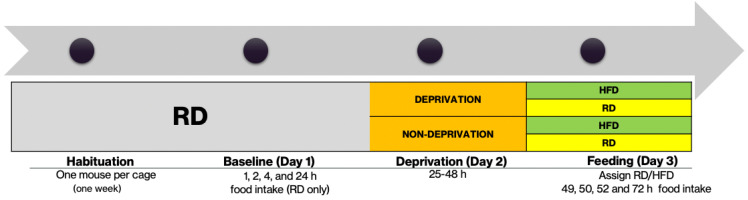
Schematic diagram of the food deprivation protocol.

**Figure 2 brainsci-13-00212-f002:**
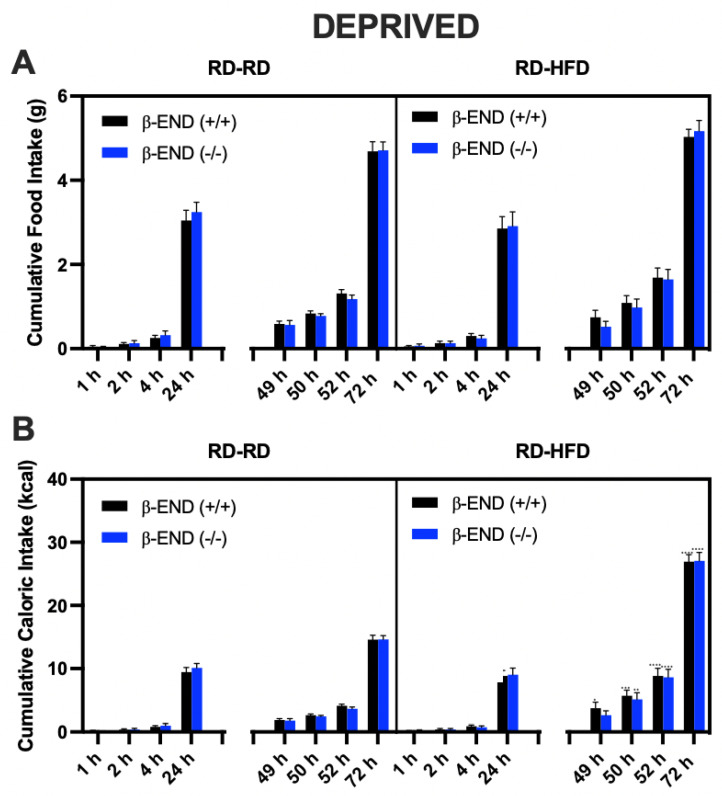
Cumulative Food consumption (**A**) and caloric intake (**B**) in deprived beta-endorphin knockout mice and their wildtype littermates/controls (*n* = 9/group). Animals were randomly assigned to receive HFD or RD following a 24 h deprivation period. Food and caloric intake were measured on day 1 (baseline: at 1, 2, 4, and 24 h) and day 3 (after deprivation: at 49, 50, 52, and 72 h). A three-way ANOVA revealed an interaction between time and food assignment in caloric intake. All data are expressed as mean ± SEM. No significant differences were found between mice of the two genotypes.

**Figure 3 brainsci-13-00212-f003:**
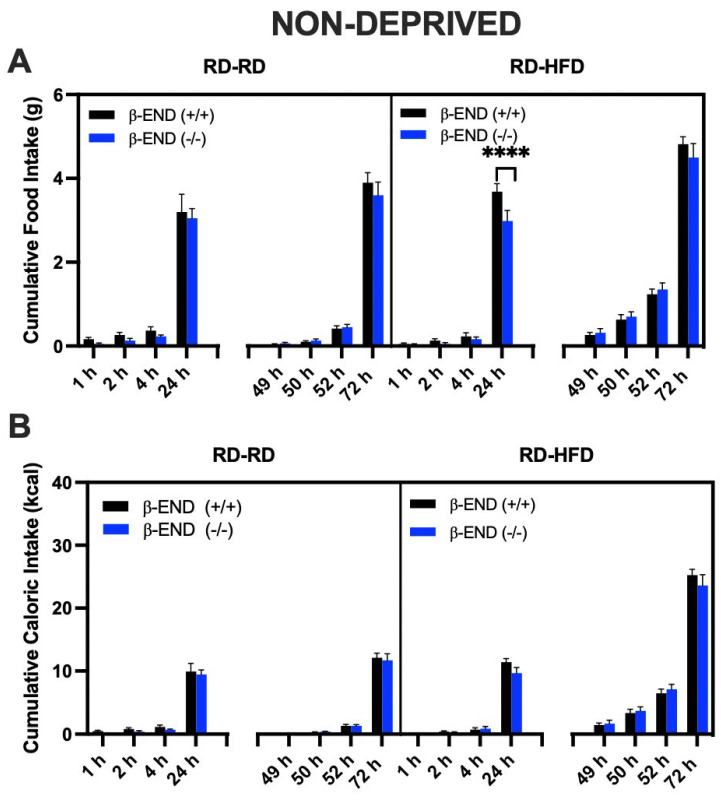
Cumulative (**A**) Food and (**B**) Caloric Intake of non-deprived beta-endorphin knockout mice and their wildtype littermates/controls (n = 6 per group). After baseline (day 2), food measurements were taken (at 1, 2, 4, and 24 h); all animals were maintained on RD for the next 24 h (day 2), then randomly assigned to receive HFD or RD for 24 h (day 3). Food and caloric intake were measured and recorded. All data are expressed as mean ± SEM. **** *p* < 0.01 by three-way ANOVA/Fisher’s LSD post hoc test.

**Figure 4 brainsci-13-00212-f004:**
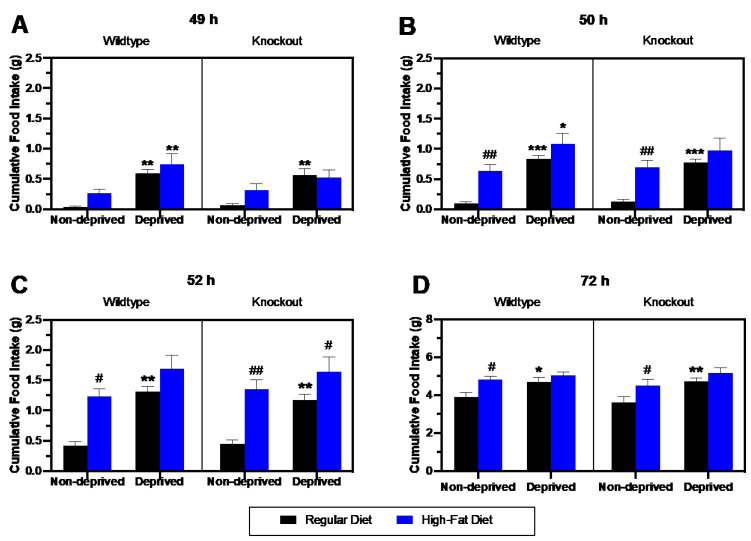
Comparison of food intake of wildtype (left half of the panel) and beta-endorphin knockout (right half) mice following 24 h food deprivation compared to their non-deprived controls at the end of the first (49 h, **A**), second (50 h, **B**), fourth (52 h, **C**) and twenty-fourth (72 h, **D**). * *p* < 0.05, ** *p* < 0.001, and *** *p* < 0.0001 signify a difference between deprived and non-deprived mice; # *p* < 0.05, and ## *p* < 0.01 signify a difference between HFD vs. RD group.

**Figure 5 brainsci-13-00212-f005:**
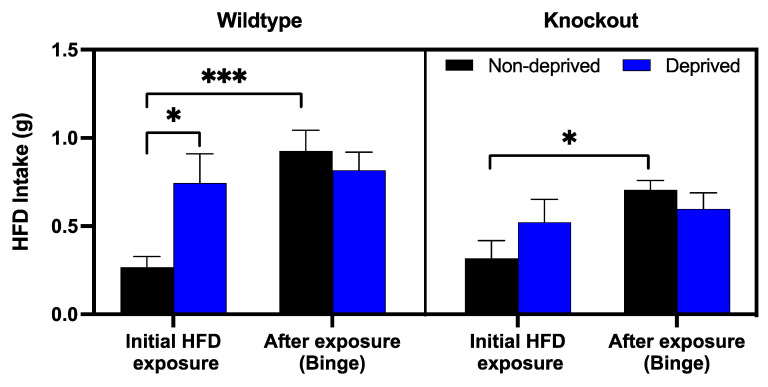
HFD consumption in non-deprived and deprived beta-endorphin knockout mice and their wildtype littermates/controls during the test for binge-eating. Food intake (g) was measured in non-deprived and deprived mice exposed to HFD for one hour and compared to their initial HFD intake, as measured in Figure 2 (deprived group) and Figure 3 (non-deprived group). All data are expressed as mean ± SEM of 6-9 mice per group. * *p* < 0.05; *** *p* < 0.0001, mixed-effect analysis/Fisher’s LSD.

## Data Availability

Data presented in this study are stored in the OneDrive folder and will become available upon reasonable request.

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
