# Peer review of "The Role of Beta-Endorphin in Food Deprivation-Mediated Increases in Food Intake and Binge-Eating"

_brainsci, 2023, doi:10.3390/brainsci13020212_

Round 1
Reviewer 1 Report
The present study evaluated the role of endogenous beta-endorphins in food-deprivation-induced increases in food intake and binge eating. Authors report that while food deprivation increased food intake, beta-endorphins may not be involved in this process. Furthermore, the authors suggested a modest role of beta-endorphins in regulating palatability-driven feeding.
I have several concerns, as listed below.
Introduction:
Line 43: Provide appropriate citations. Also, briefly discuss what authors mean by highly variable, as studies mentioned later in the paragraph support increased beta-endorphins during acute deprivation (<48hrs) conditions.
Line 62-64: For clarity, separate this complex statement into two sentences.
Methods:
Line 65: While authors rationalized 24hr time point in the food deprivation paradigm, it is unclear how one day HFD exposure and one-month later time point was chosen for the binge paradigm.
Line 81 & 94: Authors mentioned that all feeding experiments were conducted during the animal’s subjective light cycle. What was the rationale? Any possible differences if the same studies were done in rodent’s subjective dark cycle and measurements taken around shift in light/dark cycle?
Results:
Line 122: Needs revision for clarity. Also, how this comparison/statement was evaluated?
Line 126 & 149: Clearly specify in the beginning which figure panel represents food vs. caloric intake in the main text.
Line: 133, 140: Make sure to specify that it is “food-“deprived animals and several other places.
Line: 148: Provide statistical details for this statement and also show this in the figures with appropriate statistical symbols.
Line 153-154: As mentioned in Figure 1, for the first 24hr, during baseline, all animals had RD. These results basically signify individual differences/variability in groups assignments with unclear significance.
Line 160: “After baseline (day2)………………..next 24hr (day 2)”.. It does not make sense…Also, what is presented in the figure at 1,2,4 and 24 data points? Baseline or Day 2?
Line 173-175 & 221: What are first, second, fourth and twenty-fourth?
Line 225: Please define the initial exposure. It is assumed that it is the same data presented in Figures 2 and 3.
Discussion
Line 274: Please define single here
Line 273-276: here, a trend is reported, but the figure shows one star. For accurate comparison or to justify the use of the word modest – compare % change data between genotypes.
Line 276: Did the authors run the power analysis? It could be true that this study is underpowered, and an appropriate sample size is needed to make any conclusions.
Why only male mice?
Any comments if a high sugar diet was used instead of HFD?
FIGURES 2 & 3
Show all legends presented in each panel of the figure.
Since there is no sex comparison at all, the text male can be dropped from legends.
Y axis plots cumulative intake, but it could be confusing. For clarity, a break in the X-axis can be introduced, separating the first 1-24 hrs from 49-72 hrs.
Figure 3, panel A, RD-HFD: It is unclear what the statistical symbol at 24hr time point signifies.
It is unclear why beta-endorphin KO animal’s data bars are different between Fig 2 and 3.
FIGURE 4
None of the statistical symbols are defined in the figure caption.
Why food intake in gm but not calories presented?
FIGURE 5
Was there any overall difference in HFD intake between Wildtype and Knockout?
Author Response
We thank the editor and the reviewers for their timely and insightful reviews. We also thank the reviewers for their positive comments and critical review. We have modified our manuscript according to the comments of the reviewers. Below, please find our point-by-point responses to the comments of each reviewer (highlighted in the revised version of the manuscript). We hope our revised manuscript meets the requirement to be considered acceptable to your prestigious journal.
Please let me know if we can be of any further assistance.
Sincerely,
Kabirullah Lutfy, Ph.D.
Reviewer 1
Introduction:
Line 43: Provide appropriate citations. Also, briefly discuss what authors mean by highly variable, as studies mentioned later in the paragraph support increased beta-endorphins during acute deprivation (<48hrs) conditions.
Response: We have modified this sentence to address this issue. The citations include references 12-15, which following this statement (lines 42-52 of our revised manuscript).
Line 62-64: For clarity, separate this complex statement into two sentences.
Response: We have modified this sentence to address this issue (lines 59-62).
Methods:
Line 65: While authors rationalized 24hr time point in the food deprivation paradigm, it is unclear how one day HFD exposure and one-month later time point was chosen for the binge paradigm.
Response: We have addressed this issue in our revised manuscript (lines 64-67).
Line 81 & 94: Authors mentioned that all feeding experiments were conducted during the animal’s subjective light cycle. What was the rationale? Any possible differences if the same studies were done in rodent’s subjective dark cycle and measurements taken around shift in light/dark cycle?
Response: We conduct majority of our studies during the light phase of the animals but as the reviewer pointed out the response might have been different if these studies had been conducted during the dark cycle since the animals eat more during their dark cycle. However, given we did not observe much impact of beta-endorphin after deprivation when animals eat more, we may not observe a difference. This information is added to the Discussion section of our revised manuscript (lines 313-318).
Results:
Line 122: Needs revision for clarity. Also, how this comparison/statement was evaluated?
Response: There was a significant effect of time and each time point before deprivation was compared to the same time point after deprivation. This is highlighted in our revised manuscript (lines 128-129).
Line 126 & 149: Clearly specify in the beginning which figure panel represents food vs. caloric intake in the main text.
Response: This has been modified, as suggested in the paragraph (lines 124-126 and lines 148-149).
Line: 133, 140: Make sure to specify that it is “food-“deprived animals and several other places.
Response: We modified this section to address this issue (line 138).
Line: 148: Provide statistical details for this statement and also show this in the figures with appropriate statistical symbols.
Response: We have modified the graph and included the statistical details in the paragraph (148-152).
Line 153-154: As mentioned in Figure 1, for the first 24hr, during baseline, all animals had RD. These results basically signify individual differences/variability in groups assignments with unclear significance.
Response: The data are consistent in both studies when animals had regular diet intake during the first 24 h (please compare the baseline data between deprived and non-deprived groups as well as between mice of the two genotypes). However, as reviewer pointed out, this was only variable in one group out of the four groups (Fig. 3, non-deprived animals that were assigned to HFD, non-deprived RD-HFD group). We believe this is due to random sampling, but this still in the same range of baseline food intake compared to the other 3 groups (RD-RD deprived and non-deprived groups as well as deprived RD-HFD group). Only in this case, we observed a difference between wildtype and knockout mice (Fig. 3) but not other cases.
Line 160: “After baseline (day2)………………..next 24hr (day 2)”.. It does not make sense…Also, what is presented in the figure at 1,2,4 and 24 data points? Baseline or Day 2?
Response: We measured food intake for the first 24 h (day 1, baseline), then deprived the animals or left them non-deprived (day 2 or 25-48 h) and then tested them for the next 24 h (49-72 h). We have modified this in the text and in Figure 1.
Line 173-175 & 221: What are first, second, fourth and twenty-fourth?
Response: These are hours after the 24-h deprivation period. We have corrected these in the revised manuscript or included the corresponding hour, such as first (49 h), 2nd (50 h), etc.
Line 225: Please define the initial exposure. It is assumed that it is the same data presented in Figures 2 and 3.
Response: The first measurement is when animals were exposed to food during the first hour (49 h) (Figs. 2 and 3). We have clarified this in our revised manuscript (lines 231-235).
Discussion
Line 274: Please define single here
Response: We meant the initial 24-h HFD exposure (Figures 2 and 3). We have modified this in our revised manuscript. We stated this initial 24 h exposure to HFD (line 62).
Line 273-276: here, a trend is reported, but the figure shows one star. For accurate comparison or to justify the use of the word modest – compare % change data between genotypes.
Response: The trend is between wt vs ko in terms of binge eating. The difference between initial vs. second exposure was significantly different in wt (P<0.001) and ko (P<0.05) mice. We have included this information in our revised manuscript.
Line 276: Did the authors run the power analysis? It could be true that this study is underpowered, and an appropriate sample size is needed to make any conclusions.
Response: We agree with the reviewer that our study may be underpowered. However, this was part of Laica’s thesis project and now she has completed her thesis. We have included this as one of the limitations of the current study (lines 302-308).
Why only male mice?
Response: We have included this also as another limitation of the current study (lines 327-336).
Any comments if a high sugar diet was used instead of HFD?
Response: This is a great question, but we do not know if beta-endorphin would be involved in sugar intake under the conditions we used in the current study. We have conducted some operant conditioning studies using beta-endorphin wt and ko mice, but the result was not different. However, we did not assess the impact of deprivation or binge eating in those mice.
FIGURES 2 & 3
Show all legends presented in each panel of the figure.
Response: These have been changed.
Since there is no sex comparison at all, the text male can be dropped from legends.
Response: We have removed male from the legends and most of the text if we find it unnecessary.
Y axis plots cumulative intake, but it could be confusing. For clarity, a break in the X-axis can be introduced, separating the first 1-24 hrs from 49-72 hrs.
Response: The X-axis of each graph was modified as suggested.
Figure 3, panel A, RD-HFD: It is unclear what the statistical symbol at 24hr time point signifies.
Response: There a difference between wt and ko mice at that time point. We included this information in the figure legend.
It is unclear why beta-endorphin KO animal’s data bars are different between Fig 2 and 3.
Response: We have modified that.
FIGURE 4
None of the statistical symbols are defined in the figure caption.
Response: This information is included in our revised manuscript.
Why food intake in gm but not calories presented?
Response: We can but given we did not observe a significant difference between the two genotypes, we did not convert the caloric intake to per gm body weight of the animals.
FIGURE 5
Was there any overall difference in HFD intake between Wildtype and Knockout?
Response: No overall HFD intake difference was found between wildtype and knockout mice (P>0.05)

Reviewer 2 Report
ROUND 1
I would like to thank you for considering me as a reviewer in your esteemed journal; Brain Sciences is a reference for me.
Firstly I would like to inform that I don´t have any potential conflict of interest neither any other ethical concerns with regards to the paper:
- The role of beta-endorphin in food deprivation mediated increase in food intake and binge-eating.
However, at this point, I would like to advise the editor of my clinical orientation and this is more of a basic research paper (maybe I didn't notice it when I skimmed through it before accepting the review) but, the role of endorphins in eating disorders has always been of interest as it may be a potential therapeutic target. In any case, and although I am not the best person to assess the impact of this research, serious and well-done papers share some characteristics that make them easily identifiable. It was a pleasure to review this paper as the authors have shown an expertise and thoroughness that should be appreciated; the article is not riddled with grammatical, style or other mistakes, and in general it is interesting and well-written.
My major concern about this research is that for small samples (n<30) it is often difficult to detect deviations and that statistical significance does not mean too much when comparing numerical analytical values. Most of the time I review articles in which the statistical significance is not accompanied by a minimal clinical repercussion but in this case the authors do not state any clinical objective so it is not a problem
At first, the presentation of the results seemed rather poor to me (few tables without too much detail...) but it seems honest and without superfluous detours and the main finding is clear: a (very) modest role of endogenous beta-endorphin in binge eating and apart from this there is not much more to say:
- Remove please the Supplementary Materials and Appendix section and review (there is a sentence left) in the Funding section
- In the explanation of the figures (Results -lines 135 to 212-) the authors committed pseudoprecision (appearance of precision, due to many decimal places).
- I understand the article but, as I am not a native English speaker; it looks good but I don't feel qualified to judge about the English language and style.
- I have not checked the paper for plagiarism.
- I have not detected citation (or any others) mistakes in bibliography except (and this has surprised me) how outdated the bibliography is; the most recent quotation is from 2020 -and there is only one- (the rest are older).
Author Response
We thank the editor and the reviewers for their timely and insightful review. We also thank the reviewers for their positive comments and critical review. We have modified our manuscript according to the comments of the reviewers. Below, please find our point by point responses to the comments of each reviewer (highlighted in the revised version of the manuscript). We hope our revised manuscript meets the requirement to be considered acceptable to your prestigious journal.
Please let me know if we can be of any further assistance.
Sincerely,
Kabirullah Lutfy, Ph.D.
Reviewer 2
It was a pleasure to review this paper as the authors have shown an expertise and thoroughness that should be appreciated; the article is not riddled with grammatical, style or other mistakes, and in general it is interesting and well-written.
Response: We thank the reviewer for his/her positive comments.
My major concern about this research is that for small samples (n<30) it is often difficult to detect deviations and that statistical significance does not mean too much when comparing numerical analytical values. Most of the time I review articles in which the statistical significance is not accompanied by a minimal clinical repercussion but in this case the authors do not state any clinical objective so it is not a problem.
Response: We have stated the low sample size to be a limiting factor. However, as the reviewer stated, we have not made any huge claim about our data (lines 302-308).
At first, the presentation of the results seemed rather poor to me (few tables without too much detail...) but it seems honest and without superfluous detours and the main finding is clear: a (very) modest role of endogenous beta-endorphin in binge eating and apart from this there is not much more to say:
Response: We thank the reviewer for appreciating our honesty in terms of results presentation.
- Remove please the Supplementary Materials and Appendix section and review (there is a sentence left) in the Funding section
Response: We have modified these sections as suggested.
- In the explanation of the figures (Results -lines 135 to 212-) the authors committed pseudoprecision (appearance of precision, due to many decimal places).
Response: We have modified those to 3 decimal points in our revised manuscript.
- I understand the article but, as I am not a native English speaker; it looks good but I don't feel qualified to judge about the English language and style.
Response: Thank you for being honest.
- I have not checked the paper for plagiarism.
Response: Thank you for your honesty.
- I have not detected citation (or any others) mistakes in bibliography except (and this has surprised me) how outdated the bibliography is; the most recent quotation is from 2020 -and there is only one- (the rest are older).
Response: We agree with the reviewer that most of the references are in the 90s and early 2000s. However, there is not much work related to the present topic to cite in the manuscript. We found only one more article published in 2021 and added that to the list of the reference and cited that in the Introduction and Discussion (Citation # 16).
